# Exploring the Potential of vis-NIR Spectroscopy as a Covariate in Soil Organic Matter Mapping

Meihua Yang [1] , Songchao Chen [2,3] , Xi Guo [4], Zhou Shi [3] and Xiaomin Zhao [4],*

1 Department of Environmental Management, Yuzhang Normal University, Nanchang 330103, China; 0015862@zju.edu.cn
2 ZJU-Hangzhou Global Scientific and Technological Innovation Center, Hangzhou 311200, China
3 Institute of Agricultural Remote Sensing and Information Technology Application, College of Environmental and Resource Sciences, Zhejiang University, Hangzhou 310058, China; shizhou@zju.edu.cn
4 College of Land Resources and Environment, Jiangxi Agricultural University, Nanchang 330045, China; guoxi@jxau.edu.cn
* Correspondence: zhaoxm889@jxau.edu.cn

**Abstract:** Robust soil organic matter (SOM) mapping is required by farms, but their generation requires a large number of samples to be chemically analyzed, which is cost prohibitive. Recently, research has shown that visible and near-infrared (vis-NIR) reflectance spectroscopy is a fast and accurate technique for estimating SOM in a cost-effective manner. However, few studies have focused on using vis-NIR spectroscopy as a covariate to improve the accuracy of spatial modeling. In this study, our objective was to compare the mapping accuracy from a spatial model using kriging methods with and without the covariate of vis-NIR spectroscopy. We split the 261 samples into a calibration set (104) for building the spectral predictive model, a test set for generating the vis-NIR augmented set from the prediction of the fitted spectral predictive model (131), and a validation set (26) for evaluating map accuracy. We used two datasets (235 samples) for Kriging: a laboratory-based dataset (Ld, observations from calibration and test datasets) and a laboratory-based dataset with vis-NIR augmented predictions (Au.p, observations from calibration and predictions from test dataset), a laboratory-based dataset with vis-NIR spectra as the covariance (Ld.co) and augmented dataset with predictions using vis-NIR with vis-NIR spectra for the covariance (Au.p.co). The first one to seven accumulated principal components of vis-NIR spectra were used as the covariates when we used the measurement of Ld.co and Au.p.co. The map accuracy was evaluated by the validation set for the four datasets using Kriging. The results indicated that adding vis-NIR spectra as covariates had great potential in improving the map accuracy using kriging, and much higher accuracies were observed for Ld.p.co (RMSE of 5.51 g kg$^{-1}$) and Au.p.co (RMSE of 5.66 g kg$^{-1}$) than without using vis-NIR spectra as covariates for Ld (RMSE of 7.12 g kg$^{-1}$) and Au.p (RMSE of 7.69 g kg$^{-1}$). With a similar model performance to Ld.p.co, Au.p.co can reduce the cost of laboratory analysis for 60% of soil samples, demonstrating its advantage in cost-efficiency for spatial modeling of soil information. Therefore, we conclude that vis-NIR spectra can be used as a cost-effective technique to obtain augmented data to improve fine-resolution spatial mapping of soil information.

**Keywords:** soil organic matter; vis-NIR spectroscopy; mapping; covariate; ordinary kriging

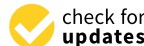



## 1. Introduction

Farmers, public policy managers, and environmental and agricultural scientists all need digital soil maps to inform appropriate decision-making for land management. Digital soil mapping (DSM) requires laboratory soil property measurements, which are costly and time-consuming. Furthermore, DSM with a high resolution requires many representative soil samples, which is costly [1,2]. Practitioners are intending to reduce the number of samples, but this may degrade the accuracy of soil maps. Balancing the budget and accuracy is a key issue for precision agriculture [3].

Visible and near-infrared (vis-NIR) spectroscopy was used as an alternative technique for laboratory chemical analysis and, therefore, may solve this problem [4]. The vis-NIR technique has received much attention during the past three decades in soil surveys and assessment studies [5], and its benefits have been documented extensively [6–8]. Several important properties of soil samples can be estimated from their scanning of samples, which is cheaper and faster than conventional laboratory methods. From this point, vis-NIR can be used as the method in DSM to solve the question of budget and sampling density [9–11].

A general method in DSM with vis-NIR spectra was to add spectral predicted soil properties to the laboratory soil measurements, called the augmented data, to increase sampling density [1]. For example, Viscarra Rossel et al. [12] showed that augmented data could improve the accuracy of soil maps. DSM with augmented data was carried out by spatial interpolation, predominantly by kriging. Kriging requires the underlying random variable to be approximately normally distributed, but the estimated properties from vis-NIR spectral predictive models can be inaccurate and skewed [13]. Furthermore, spectral predicted soil properties, compared with laboratory soil measurements, tend to easily smooth the variation [14]. Therefore, the accuracy is frequently unsatisfactory in soil mapping with vis-NIR augmented data using kriging methods [15].

Another general method in DSM with vis-NIR spectra is to use the predicted value from the nonlinear model (such as Cubist, Random Forest) using spectra and the environmental covariates [16,17]. These studies can tell us the usefulness of vis-NIR for mapping but do not show how much the extent of vis-NIR can improve.

How to improve accuracy and reduce cost when using vis-NIR spectroscopy requires studies focused on methods to reduce the negative impacts of modeling error [2,5]. Obtaining better covariates that can capture the soil formation factors could be useful in improving the accuracy [2]. Given that some of the key aspects listed above previously existed in mapping using vis-NIR spectroscopy, our study aimed to (i) demonstrate the potential for using vis-NIR data for DSM at a field scale and (ii) compare the accuracy of maps based on laboratory measurement and the accuracy with those using a vis-NIR augmented dataset with and without using vis-NIR spectra as the covariate.

In this study, we used four kinds of data sources for mapping to explore the potential of using vis-NIR spectra for mapping: (1) laboratory-based dataset (Ld, observations from calibration and test datasets); (2) the sum of data from a laboratory-based dataset from calibration and predicted SOM data using vis-NIR calibration model and vis-NIR spectra from the test dataset, which called vis-NIR augmented data (Au.p); (3) laboratory-based dataset with vis-NIR spectra from the total dataset as the covariance (Ld.co); (4) augmented dataset with vis-NIR spectra from the total dataset as the covariance (Au.p.co). We also analyzed the first one to seven accumulated principal components of vis-NIR spectra, which were used as the covariates when we used the measurement of Ld.co and Au.p.co. We also discussed four preprocessing for vis-NIR spectra when Au.p, Ld.co, Au.p.co.

## 2. Materials and Methods

### 2.1. Study Area and Soil Sampling

The study area is located in Ji'an County, eastern Jiangxi Province (Figure 1), and covers 2470 ha at altitudes ranging from 50 to 60 m above sea level. The location is distributed on both sides of the tributaries of the Ganjiang River with flat terrain. The predominant soil types came from river alluvials. This area is used for high-quality prime farmland [18]. Soil texture varies from loamy sand to clay. We collected 261 samples on a regular grid of 300 × 300 m from the arable layer (0–20 cm) (Figure 1). In this study, we used soil organic matter (SOM), which can be documented to be accurately estimated by vis-NIR.

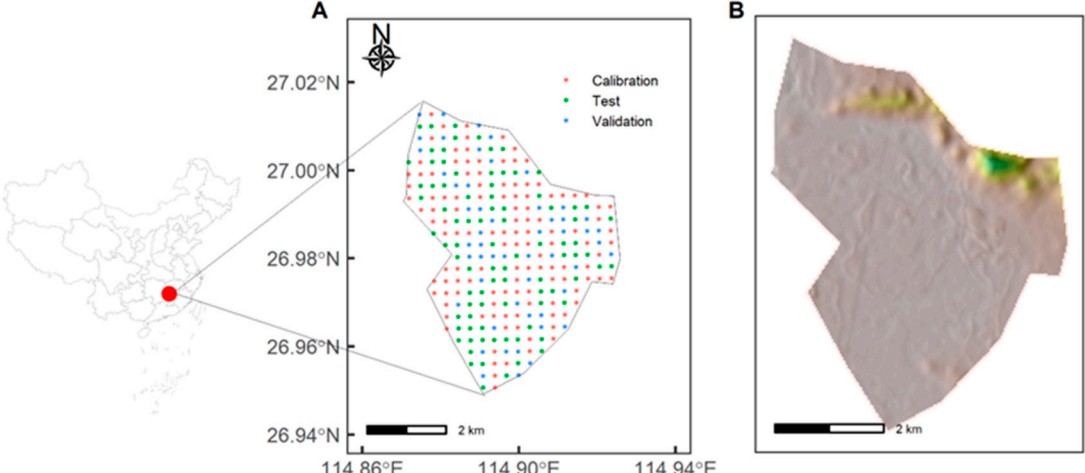

**Figure 1.** The location of sampling (**A**) and the DEM of the sampling location (**B**).

### 2.2. Measurement and Processing of vis-NIR Spectra

After soil samples were collected at the locations from the surface soil layer, they were mixed into a composite sample before being stored in a labeled plastic bag. All composite samples were transported to the laboratory, air-dried, ground, and sieved to pass through a 2 mm sieve. Each sample was divided into two portions using the quartering method, one for laboratory chemical analysis and the other for spectral measurements. For the dry spectral measurements, each soil sample was placed in a Petri dish with a diameter of 10 cm and a depth of 1.5 cm. Soil spectra were measured using the same ASD FieldSpec4 spectrometer (350–2500 nm) equipped with a contact probe (Malvern Panalytical Ltd., Malvern, UK). The spectra were taken at three arbitrarily selected locations from the soil surface. Ten spectra were recorded in each of the three sensing locations. The thirty total spectra were averaged to one to represent the spectra of the soil sample. The spectra in the range of 400 to 2400 nm were used as the final spectra for the next use.

The vis-NIR reflectance spectra were transformed to apparent absorbance (log10 1/reflectance). Then, we applied the Savitzky–Golay (window size = 11, polynomial order = 3, differentiation order = 1) filter (lg_sg), [19] standard normal variate (lg_snv) [20], multiplicative scatter correction + standard normal variate (msc_snv) and detrend normalization (lg_dt) to compare which processing was optimal. We used the 400–2450 nm spectral range to eliminate the high signal noise at the two ends of the spectrometer. The total spectral bands of 2051 were used.

Samples were randomly split into calibration (104, 40%), validation (26, 10%), and test (31, 50%) sets.

The samples were packed into plastic bags, labeled, and transported to the laboratory. The soil samples were air-dried, ground, and sieved to less than 2 mm. Stones and plant residues were removed.

### 2.3. Spectroscopic Model and Augmented Data

We used PLSR [21] as the spectroscopic model. PLSR is a linear regression model that is widely used for spectroscopic soil modeling. The optimum number of latent variables for PLSR was determined by the lowest root mean square error (RMSE) using leave-one-out cross-validation.

The predicted SOM from the PLSR model using the calibration datasets whose spectra with four processing steps (see above) were pooled to the SOM data from the calibration, which is called the augmented data, resulting in four augmented datasets.

### 2.4. Laboratory Analyses of Soil Organic Matter (SOM)

SOC content was measured using the $H_2SO_4$-$K_2Cr_2O_7$ oxidation method at 180 °C for 5 min method according to the methods of the Institute of Soil Science of the Chinese Academy of Sciences [22]. The SOM content was obtained using SOC multiplied by the coefficient of 1.72, which was suggested by Reference [23].

### 2.5. Selection of Covariates

We analyzed the correlation of SOM with the terrain attributes (terrain roughness, vector terrain roughness, slope of aspect, slope of slope), but the correlations were quite low, so we did not consider them. We only used the spectra as the covariate. When using vis-NIR spectroscopy as the covariate, those PCs on the scaled spectra with significant $p$ values ($p < 0.01$) with SOM were selected. PCs with $p$ values that were negative were transformed into negative PCs, which ensured that the correlation between the PCs and SOM was positive. The spectra used as the covariate were the spectra from the total samples under four spectral preprocessing steps (Figure 2).

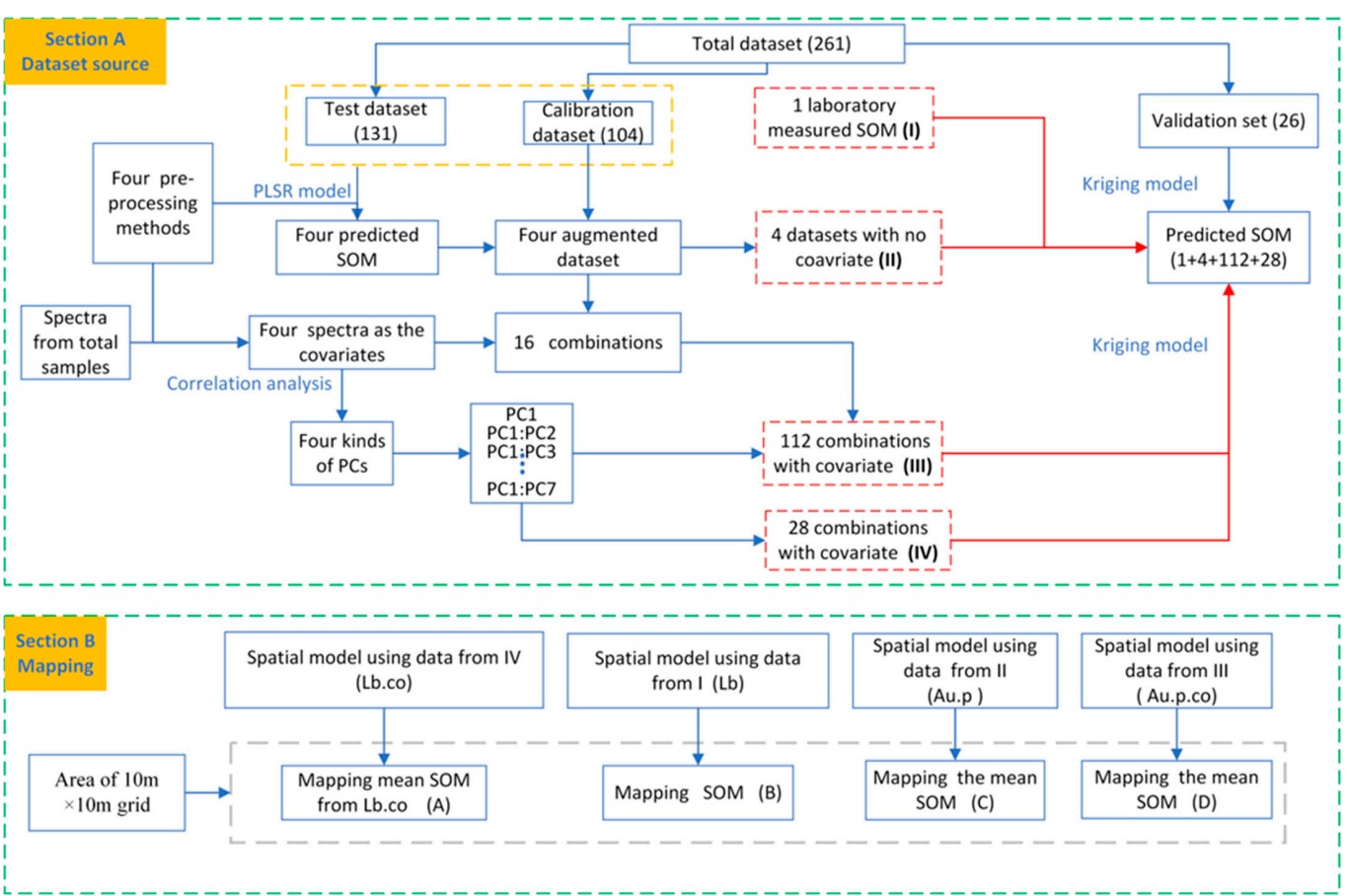

**Figure 2.** Dataset from the combination and the mapping flowchart. The letters (A)–(D) correspond with data of the laboratory-based (Lb), vis-NIR augmented datasets (Au.p), the laboratory-based with pcs covariate (Ld.co), vis-NIR augmented datasets with pcs covariate (Au.p.co).

### 2.6. Spatial Modeling, Performance Estimation, and SOM Mapping

In this study, area and soil properties (i.e., SOM) at location s were defined as:

$$Y_i = u_i + Z_i(s) + \varepsilon(s) \tag{1}$$

where $u_i$ was a mean, $Z_i(s)$ was normally distributed, and $\varepsilon(s)$ was an uncorrelated random error.

Autocorrelated spatially random component with zero mean, unit variance, and variogram:

$$\gamma(h) = \frac{1}{2}var[\varepsilon(\boldsymbol{u}) - \varepsilon(\boldsymbol{u} + \boldsymbol{h})]] = \frac{1}{2}E[]\{\varepsilon(\boldsymbol{u}) - \varepsilon(\boldsymbol{u} + \boldsymbol{h})\}^2 \tag{2}$$

where $\varepsilon(u)$ and $\varepsilon(u + h)$ are random variables at places $u$ and $u + h$ separated by the vector h, and E denotes the expectation.

The semivariance models were spherical functions, which can be defined below:

$$\gamma(h) = \begin{cases} \tau + \sigma\left\{ \frac{3h}{2\alpha} + \frac{1}{2}\left(\frac{h}{\alpha}\right)^3 \right\} & h \le \alpha \\ \tau + \sigma & h > \alpha \\ 0 & h = 0 \end{cases} \tag{3}$$

where $\tau$ is the nugget variance, which can be attributed to measurement errors or spatial sources of variation within the range of the sampling interval, and $\alpha$ is the range of spatial dependence or spatial autocorrelation; $\gamma(h)$ is the semivariance at lag $h$ and $\sigma$ is the a priori variance of the autocorrelated process.

The parameters of the autocorrelation model of $\sigma$, $\tau$, $\alpha$, the mean u, and the realizations of $Z(s)$ are unknown and must be estimated from the data. The values of $\sigma$, $\tau$, and $\alpha$ were determined by fitting a model to the points forming from the empirical semivariogram. Predictions at unvisited locations were made by global block kriging on a 10 m × 10 m grid.

To show how to improve the prediction accuracy when using the spectra as the covariate, we first calculated the results induced from the augmented data that used augmented SOM predicted from the four spectral processing methods. Then, we calculated the results from the combination of the four kinds of augmented data and four preprocessed spectra as covariates. When considering the covariate, we calculated 1 to 7 principal components from each preprocessing technique. Therefore, the total predicted results of 116 (112 predicted SOM using covariance when using the augmented data and 4 augmented data with no covariate, see Figure 2B, Section A).

To further compare which measure made the main effect in predicting and mapping, we also mapped the SOM results using geographic models from the library-based data with the PCs as the covariate using lg_dt preprocessing, which was called Lb.co. The mapping that was finally shown was from the laboratory-based (Ld), laboratory-based with PCs as the covariate using lg_dt pre-proposing (Lb.co), vis-NIR augmented data using whose spectra using lg_dt pre-proposing (Au.p) and vis-NIR augmented data without PCs as the covariate whose all spectra used lg_dt pre-possessing (Au.p.co); this part can be seen in Section B in Figure 2.

Root mean squared error (RMSE), R2, bias and the ratio of performance to interquartile range (RPIQ) were used to evaluate the accuracy and bias of prediction at the validation set for SOM.

## 3. Results

### 3.1. SOM and vis-NIR Spectra

Table 1 presents a summary of the calibration, test, and validation datasets. The mean SOM from the three datasets was similar. The skew of the calibration was greater than that of the test and validation datasets due to the largest SOM in calibration.

Apparent water absorption peaks were observed in the vis-NIR spectra bands near 1400 nm and 1900 nm, as shown in Figure 3. The absorption peaks near 2200 nm indicated the existence of kaolinite in the soil samples [19]. In addition, the absorption spectra at approximately 450 and 850 nm showed goethite and hematite, which are characteristic of iron-bearing minerals [20]. Three samples had a higher reflectance at 600–1400 nm than the rest of the soil samples because of their relatively late development [20].

**Table 1.** Summary statistics of SOM of calibration, test, and validation datasets.

| | N | Mean | SD | Skew | Min. | 1st Qu. | Median | 3rd Qu. | Max. |
|---|---|---|---|---|---|---|---|---|---|
| Calibration /g kg$^{-1}$ | 104 | 18.92 | 8.87 | 0.61 | 2.31 | 12.08 | 18.72 | 23.55 | 49.68 |
| Test/g kg$^{-1}$ | 131 | 18.32 | 6.57 | 0.31 | 2.06 | 14.37 | 16.99 | 23.31 | 35.58 |
| Validation /g kg$^{-1}$ | 26 | 17.63 | 7.36 | 0.31 | 4.96 | 12.22 | 17.59 | 23.26 | 37.09 |

n is the number of samples, SD is the standard deviation, Min. is minimum, 1st Qu. Is the first quantile, 3rd Qu. Is the third quantile, and Max. is the maximum.

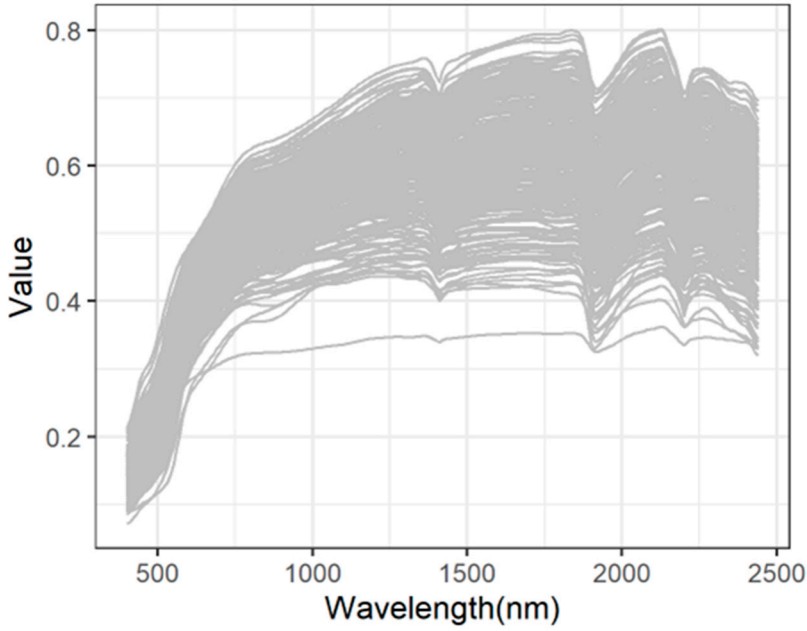

**Figure 3.** Reflectance of the samples.

*3.2. Prediction of PLSR*

The results showed that the vis-NIR spectroscopy calibration model using the 101 calibration samples had a better-predicted performance for the test dataset (RPIQ value was from 2.03 to 2.30) than for the validation dataset (RPIQ values was from 1.51 to 1.71) (Table 2). The model using the spectra with preprocessing of msc.snv and lg_dt gave the best performance with RPIQ values of 1.71 and 2.30 for the validation dataset and the test dataset, respectively. The preprocessing of msc.snv was more effective for the validation than for the test validation. The bias values showed that predictions by the vis-NIR models for validation were positively biased.

**Table 2.** Prediction results of the validation and test dataset from vis-NIR calibration models under four spectral preprocessing methods.

| Transformation | Validation Dataset | | | | Test Dataset | | | |
|---|---|---|---|---|---|---|---|---|
| | RMSE | RPIQ | Bias | R$^2$ | RMSE | RPIQ | Bias | R$^2$ |
| lg.sg | 6.11 | 1.51 | 0.11 | 0.56 | 4.32 | 2.17 | 0.39 | 0.69 |
| msc.snv | 5.40 | 1.71 | 0.09 | 0.66 | 4.61 | 2.03 | 0.20 | 0.64 |
| lg.snv | 5.74 | 1.61 | 0.71 | 0.63 | 4.19 | 2.23 | 0.31 | 0.71 |
| lg.dt | 5.48 | 1.68 | 0.63 | 0.65 | 4.07 | 2.30 | −0.04 | 0.72 |

Note: lg_sg: Savitzky-Golay of log(1/(R)), lg_snv: standardNormalVariate of log(1/(R)), msc_snv: multiplicative scatter correction + standardNormalVariate of log(1/(R)), lg_dt: detrended normalization of log(1/(R)).

### 3.3. Correlation Analysis

Figure 4A shows that the correlation of SOM with PCs calculated from the different prepossessed spectra was different. The values of correlation calculated from the first four PCs and spectra under msc_snv and lg_dt were significant. However, after msc_snv and lg_sg were proposed, the significant correlation value showed that the PCs were seven (the minimum significant correlation value was 0.18). Among the four preprocessing methods, lg_sg had a more negative correlation than the other three preprocessed methods. The following calculation involved the PCs that used the maximum of seven.

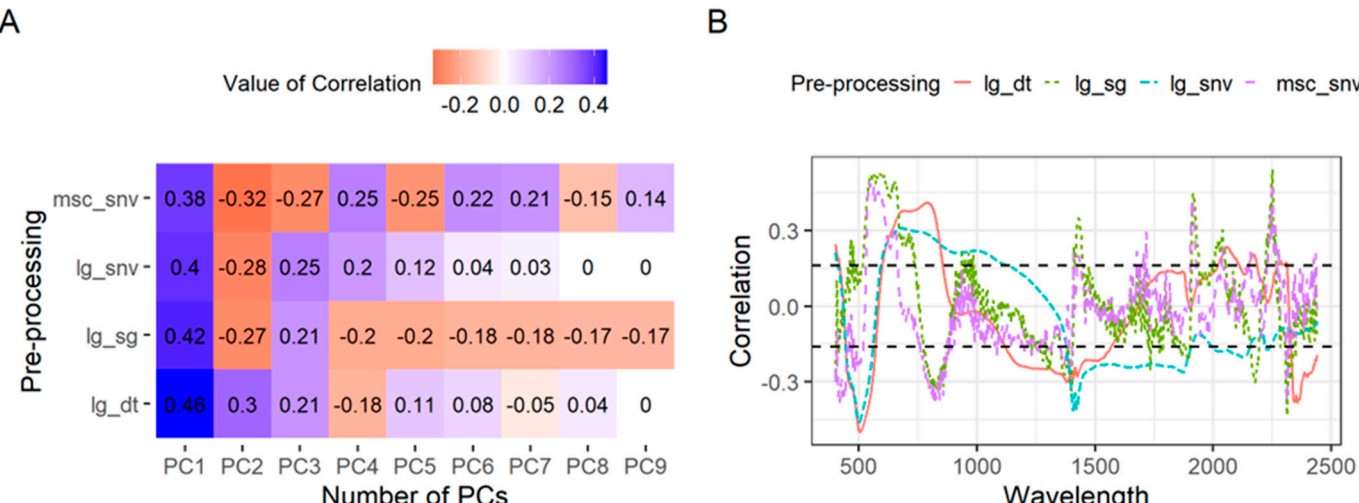

**Figure 4.** The correlation of SOM with principal components (PC1:PC9) calculated from the spectroscopy reflectance R with four preprocessing methods (**A**) and of SOM with spectra with four preprocessing methods (**B**). Note: lg_sg: Savitzky-Golay of log(1/(R)), lg_snv: standardNormalVariate of log(1/(R)), msc_snv: multiplicative scatter correction + standardNormalVariate of log(1/(R)), lg_dt: detrended normalization of log(1/(R)). Note: lg_sg: Savitzky-Golay of log(1/(R)), lg_snv: standardNormalVariate of log(1/(R)), msc_snv: multiplicative scatter correction + standardNormalVariate of log(1/(R)), lg_dt: detrended normalization of log(1/(R)). The two values from the black dotted lines are correlation values whose *p* value was significant ($p < 0.01$).

Figure 4B shows that the correlations of SOM and spectra with proposals from lg_dt and lg_snv were more significant than with proposals from lg_sg and msc_snv. The number of significant bands from four preprocessing steps was as follows: lg_snv (1248) > lg_dt (1204) > lg_sg (761) > msc_snv (584).

### 3.4. The Prediction from the Spatial Model with Spectra as the Covariate

Figure 5 shows that along each row, the change in RMSE was similar, but along each column had a difference, which showed that the effect from the spectra as the covariate was larger than the vis-NIR augmented data induced from the different preprocessing. When considering the whole from each line, the difference from lg_snv_co with different PCs was the smallest among the four lines. The results from lg_dt_co and msc_snv_co had the same trend, but the number of PCs with the lowest RMSE value was different, with the former being four and the latter being five. The RMSE from lg_sg_co showed that the lowest value was PC two.

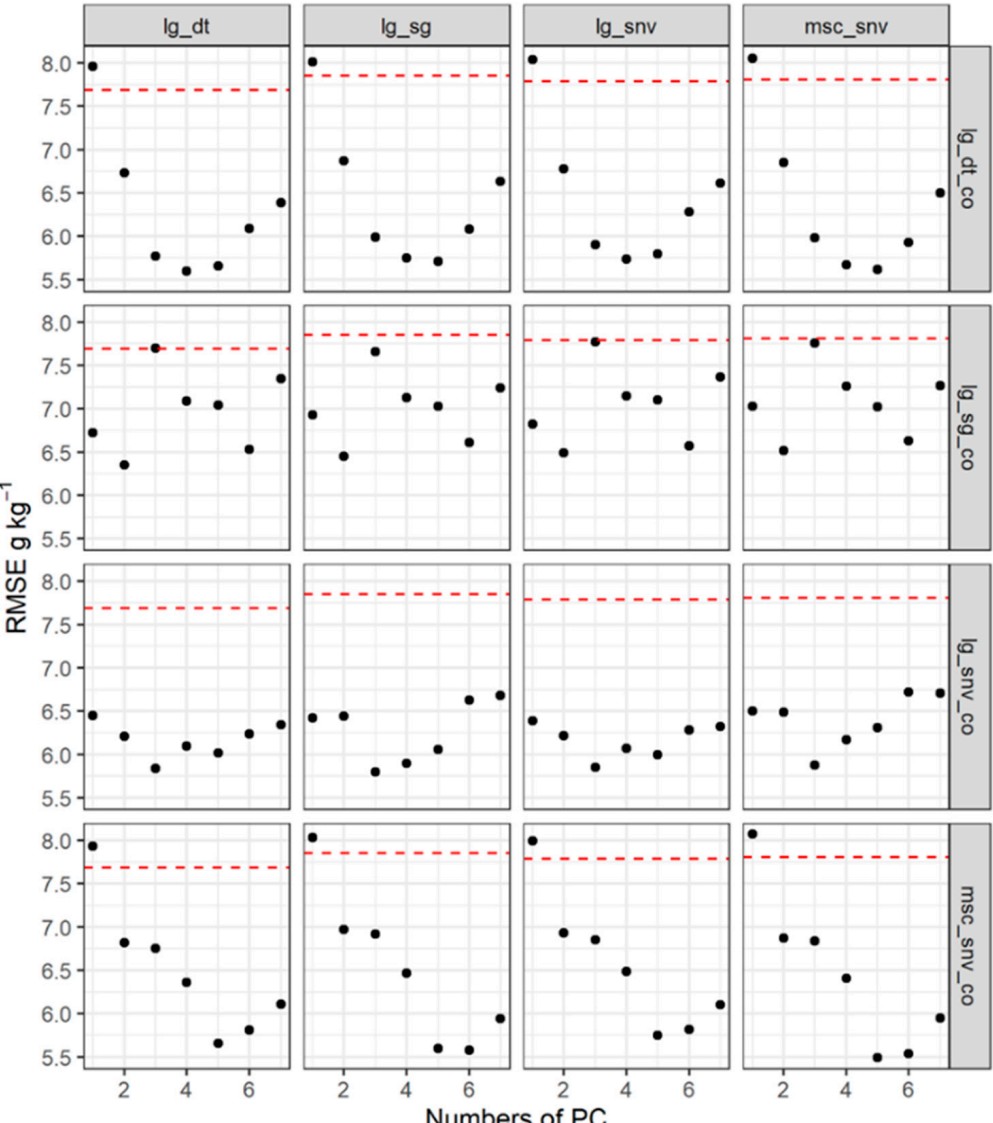

**Figure 5.** The RMSE calculated by the spatial model with four augmented data and 7 PCs as covariates induced from the spectra under four preprocessing methods. Each column represents that the augmented data were the same, but PCs (as the covariate) were induced from the different preprocessing methods. Each row representing PCs (as the covariate) was the same, but the augmented data were induced from the different preprocessing methods. The red lines show the RMSE value without the spectra as the covariate in the corresponding column, and each column had the same RMSE.

When comparing the results of RMSE from the spatial model with and without spectra as the covariate, we found that regardless of the kind of pretreatment with PCs as the covariate, the RMSE value was reduced to different degrees except for the first PC from lg_dt_co and msc_snv_co. When not using the spectra as the covariate, the effect of preprocessing on the prediction result was not obvious, and the best result was obtained from lg_dt (Figure 5, red line in each column).

The bias in Figure 6 shows that the different PCs from lg_dt_co and lg_snv_co had little effect. lg_snv_co had a bias that was lower than that without the covariate (lg_snv, Figure 6 red lines). The changes in bias from lg_dt_co were all around the bias value of lg_dt with a short distance except for the seventh PC. When the PC was 3–5, the biases from lg_sg_co and msc_snv_co were higher than those from lg_sg and msv_snv, which showed that when the spectra were used as the covariate, the reduced RMSE was not due to the reduced bias.

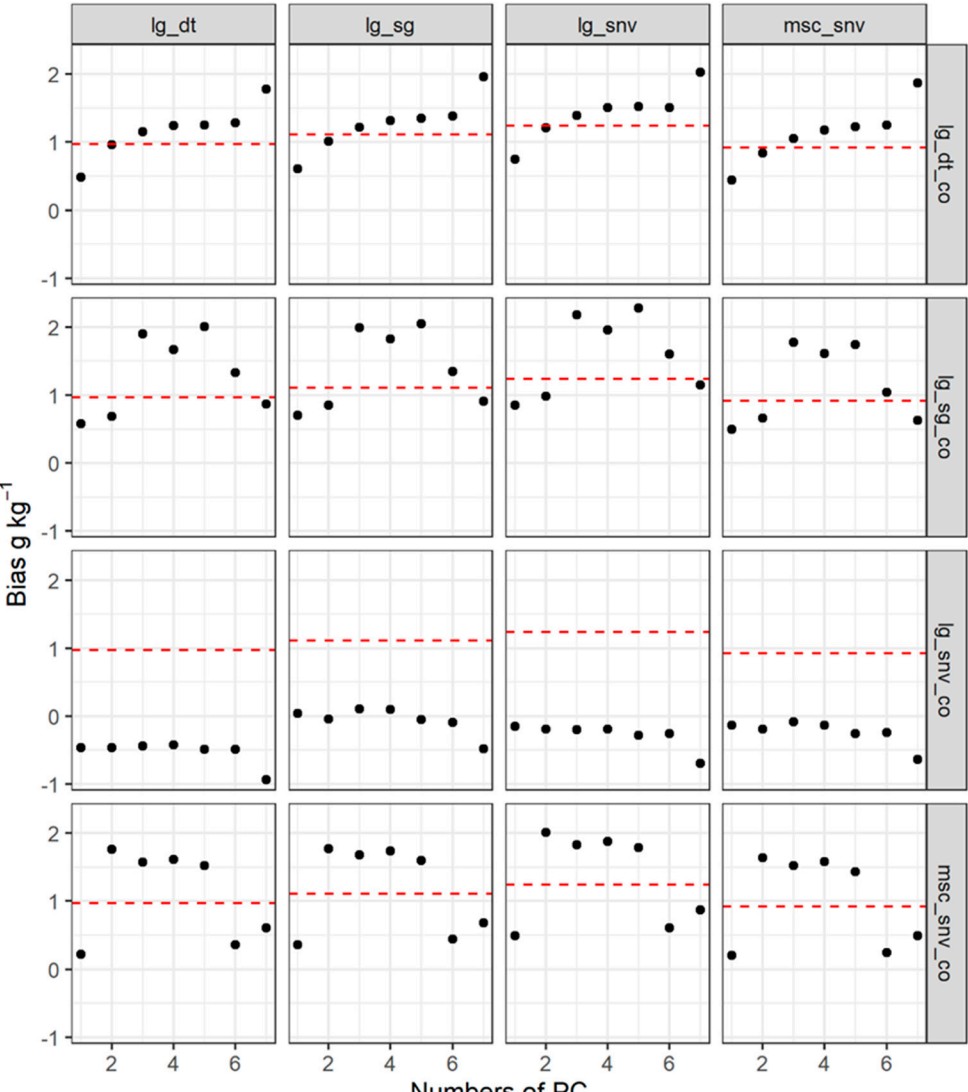

**Figure 6.** The bias calculated by the spatial model with four augmented data points and 7 PCs as covariates induced from the spectra under four preprocessing methods. Each column represents that the augmented data were the same, but PCs (as the covariate) were induced from the different preprocessing methods. Each row representing PCs (as the covariate) was the same, but the augmented data were induced from the different preprocessing methods. The red lines show the RMSE value without the spectra as the covariate in the corresponding column, and each column had the same bias.

### 3.5. Spatial Analysis and SOM Mapping

The spatial predictions using the SOM value from the vis-NIR augmented dataset (Au.p) were slightly better than those using the data from the laboratory analysis (Lb) (Tables 3 and 4) in both crossing validation and validation. When comparing the prediction results with and without covariates of vis-NIR spectra, the former obtained higher prediction accuracy (Tables 3 and 4). The model from the laboratory with vis-NIR spectra as the covariate (Lb.co) had a similar result to the model from the vis-NIR augmented dataset with vis-NIR spectra as the covariates (Au.p.co). The ME values in Table 3 show that the spatial predictions from the cross-validation were always slightly negatively biased, but the validation was always positively biased.

**Table 3.** The statistics of the cross-validation and prediction results of kriging for SOM.

| | LVs | | Cross-Validation | | | | Validation | | |
|---|---|---|---|---|---|---|---|---|---|
| | | $R^2$ | RMSE /g kg$^{-1}$ | RPIQ | ME /g kg$^{-1}$ | $R^2$ | RMSE/g kg$^{-1}$ | RPIQ | ME /g kg$^{-1}$ |
| Lb | 4 | 0.18 | 6.83 | 1.54 | 1.01 | 0.27 | 7.12 | 1.27 | 1.04 |
| Au.p | 6 | 0.18 | 6.74 | 1.56 | 0.68 | 0.35 | 7.69 | 1.20 | 0.97 |
| Lb.co | 7 | 0.50 | 5.36 | 2.08 | 0.17 | 0.58 | 5.51 | 1.64 | 0.74 |
| Au.p.co | 8 | 0.47 | 5.46 | 1.86 | 0.34 | 0.66 | 5.66 | 1.65 | 1.37 |

Ld: laboratory-based dataset, Au.p: augmented dataset, Ld.co: laboratory-based dataset with vis-NIR spectra as the covariate, Au.p.co: augmented dataset with vis-NIR spectra as the covariate. LVs: latent variable.

**Table 4.** The statistics of prediction results of kriging for SOM using different principal numbers under different preprocessing methods.

| PCs | Pre-Processing | RMSE | RPIQ | Bias | $R^2$ | RMSE | RPIQ | Bias | $R^2$ | RMSE | RPIQ | Bias | $R^2$ | RMSE | RPIQ | Bias | $R^2$ |
|---|---|---|---|---|---|---|---|---|---|---|---|---|---|---|---|---|---|
| | | | lg.sg | | | | msc.snv | | | | lg.snv | | | | lg.dt | | |
| 1 | lg.sg | 6.93 | 1.33 | 0.7 | 0.45 | 7.03 | 1.31 | 0.5 | 0.42 | 6.82 | 1.35 | 0.85 | 0.47 | 6.72 | 1.37 | 0.58 | 0.48 |
| | msc.snv | 8.03 | 1.15 | 0.36 | 0.24 | 8.07 | 1.14 | 0.2 | 0.23 | 7.99 | 1.15 | 0.49 | 0.24 | 7.93 | 1.16 | 0.22 | 0.25 |
| | lg.snv | 6.42 | 1.44 | 0.04 | 0.52 | 6.5 | 1.42 | −0.13 | 0.51 | 6.39 | 1.44 | −0.15 | 0.52 | 6.45 | 1.43 | −0.46 | 0.51 |
| | lg.dt | 8.01 | 1.15 | 0.61 | 0.25 | 8.05 | 1.15 | 0.44 | 0.24 | 8.04 | 1.15 | 0.75 | 0.24 | 7.96 | 1.16 | 0.48 | 0.25 |
| 2 | lg.sg | 6.45 | 1.43 | 0.85 | 0.54 | 6.52 | 1.42 | 0.66 | 0.52 | 6.49 | 1.42 | 0.98 | 0.53 | 6.35 | 1.45 | 0.69 | 0.54 |
| | msc.snv | 6.97 | 1.32 | 1.77 | 0.46 | 6.87 | 1.34 | 1.64 | 0.47 | 6.93 | 1.33 | 2.01 | 0.48 | 6.82 | 1.35 | 1.76 | 0.49 |
| | lg.snv | 6.44 | 1.43 | −0.04 | 0.56 | 6.49 | 1.42 | −0.19 | 0.55 | 6.22 | 1.48 | −0.19 | 0.55 | 6.21 | 1.49 | −0.46 | 0.55 |
| | lg.dt | 6.87 | 1.34 | 1.01 | 0.47 | 6.85 | 1.35 | 0.84 | 0.47 | 6.78 | 1.36 | 1.21 | 0.49 | 6.73 | 1.37 | 0.96 | 0.49 |
| 3 | lg.sg | 7.66 | 1.2 | 1.99 | 0.36 | 7.76 | 1.19 | 1.78 | 0.34 | 7.77 | 1.19 | 2.18 | 0.36 | 7.7 | 1.2 | 1.9 | 0.36 |
| | msc.snv | 6.92 | 1.33 | 1.68 | 0.46 | 6.84 | 1.35 | 1.52 | 0.47 | 6.85 | 1.35 | 1.83 | 0.48 | 6.75 | 1.37 | 1.57 | 0.49 |
| | lg.snv | 5.8 | 1.59 | 0.11 | 0.64 | 5.88 | 1.57 | −0.08 | 0.63 | 5.85 | 1.58 | −0.2 | 0.59 | 5.84 | 1.58 | −0.44 | 0.6 |
| | lg.dt | 5.99 | 1.54 | 1.22 | 0.59 | 5.98 | 1.54 | 1.05 | 0.59 | 5.9 | 1.56 | 1.39 | 0.61 | 5.77 | 1.6 | 1.15 | 0.62 |
| 4 | lg.sg | 7.13 | 1.29 | 1.83 | 0.45 | 7.26 | 1.27 | 1.61 | 0.43 | 7.15 | 1.29 | 1.96 | 0.47 | 7.09 | 1.3 | 1.67 | 0.47 |
| | msc.snv | 6.47 | 1.42 | 1.74 | 0.54 | 6.41 | 1.44 | 1.58 | 0.54 | 6.49 | 1.42 | 1.88 | 0.55 | 6.36 | 1.45 | 1.61 | 0.55 |
| | lg.snv | 5.9 | 1.56 | 0.1 | 0.6 | 6.17 | 1.5 | −0.13 | 0.55 | 6.07 | 1.52 | −0.19 | 0.56 | 6.1 | 1.51 | −0.42 | 0.56 |
| | lg.dt | 5.75 | 1.6 | 1.32 | 0.63 | 5.67 | 1.63 | 1.18 | 0.63 | 5.74 | 1.61 | 1.51 | 0.64 | 5.6 | 1.65 | 1.24 | 0.65 |
| 5 | lg.sg | 7.03 | 1.31 | 2.05 | 0.47 | 7.02 | 1.31 | 1.74 | 0.46 | 7.1 | 1.3 | 2.28 | 0.47 | 7.04 | 1.31 | 2.01 | 0.47 |
| | msc.snv | 5.6 | 1.65 | 1.6 | 0.66 | 5.49 | 1.68 | 1.43 | 0.67 | 5.75 | 1.6 | 1.79 | 0.65 | 5.66 | 1.63 | 1.52 | 0.65 |
| | lg.snv | 6.06 | 1.52 | −0.05 | 0.58 | 6.31 | 1.46 | −0.26 | 0.54 | 6 | 1.54 | −0.28 | 0.57 | 6.02 | 1.53 | −0.49 | 0.57 |
| | lg.dt | 5.71 | 1.62 | 1.35 | 0.64 | 5.62 | 1.64 | 1.23 | 0.64 | 5.8 | 1.59 | 1.52 | 0.64 | 5.66 | 1.63 | 1.25 | 0.65 |
| 6 | lg.sg | 6.61 | 1.4 | 1.35 | 0.52 | 6.63 | 1.39 | 1.04 | 0.51 | 6.57 | 1.4 | 1.6 | 0.54 | 6.53 | 1.41 | 1.33 | 0.55 |
| | msc.snv | 5.58 | 1.65 | 0.44 | 0.64 | 5.54 | 1.66 | 0.24 | 0.64 | 5.82 | 1.59 | 0.61 | 0.62 | 5.81 | 1.59 | 0.36 | 0.62 |
| | lg.snv | 6.63 | 1.39 | −0.09 | 0.5 | 6.72 | 1.37 | −0.24 | 0.48 | 6.28 | 1.47 | −0.26 | 0.54 | 6.24 | 1.48 | −0.49 | 0.55 |
| | lg.dt | 6.08 | 1.52 | 1.38 | 0.59 | 5.93 | 1.56 | 1.25 | 0.6 | 6.28 | 1.47 | 1.51 | 0.57 | 6.09 | 1.51 | 1.28 | 0.59 |
| 7 | lg.sg | 7.24 | 1.27 | 0.91 | 0.46 | 7.27 | 1.27 | 0.63 | 0.46 | 7.37 | 1.25 | 1.15 | 0.46 | 7.35 | 1.26 | 0.87 | 0.46 |
| | msc.snv | 5.94 | 1.55 | 0.68 | 0.59 | 5.95 | 1.55 | 0.49 | 0.59 | 6.1 | 1.51 | 0.87 | 0.59 | 6.11 | 1.51 | 0.61 | 0.58 |
| | lg.snv | 6.68 | 1.38 | −0.48 | 0.48 | 6.71 | 1.38 | −0.64 | 0.48 | 6.32 | 1.46 | −0.69 | 0.54 | 6.34 | 1.45 | −0.93 | 0.54 |
| | lg.dt | 6.63 | 1.39 | 1.96 | 0.54 | 6.5 | 1.42 | 1.87 | 0.54 | 6.61 | 1.4 | 2.03 | 0.56 | 6.39 | 1.44 | 1.78 | 0.57 |

Figure 7 shows the spatial predictions from the maps produced by four kinds of data. When the data were laboratory-based and vis-NIR augmented, the spatial prediction was similar, and the differences between them were relatively small. The largest difference between maps is located in the northwest part of the study area, where the altitudes are highest (Figure 1B). The main difference existed with a lower SOM content in the laboratory-based model, but a higher SOM content in the vis-NIR augmented models. Another noticeable difference was in the middle part of the study area, where the maps predicted with the vis-NIR augmented dataset were smoother than those predicted with the laboratory-based dataset. When comparing the maps from the laboratory data with and without the PC covariate (Lb.co vs. Lb), the main difference was located in the northwest, where the SOM value from Lb.co was higher than that from Lb. There were some differences between maps from Lb and Au.p.co in the middle of sampling and northwest, where the SOM value from Au.p.co was lower than that from Lb.

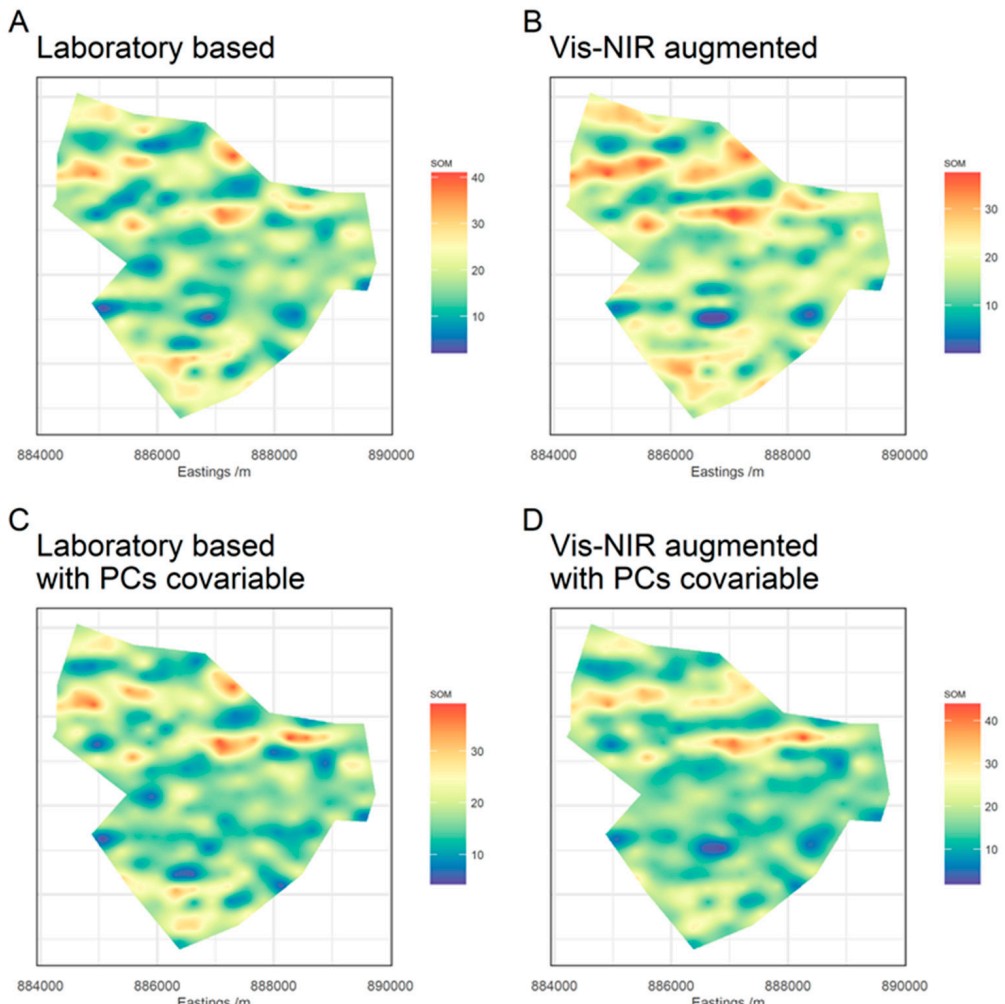

**Figure 7.** Maps of SOM generated from the data of the laboratory-based (Lb) (**A**), vis-NIR augmented datasets (Au.p) (**B**), the laboratory-based with pcs covariate (Ld.co) (**C**), vis-NIR augmented datasets with pcs covariate (Au.p.co) (**D**).

The histogram showed that the percent of predicted SOM in all ranges between the maps from the data of Lb and Au.p had no difference (Figure 8). However, when considering the vis-NIR spectra as covariates, the histograms from Ld.co and Au.p.co in the SOM content range of 12–21 g kg$^{-1}$ had a larger percentage (58%) than Lb and Au.p (64%), which can also be seen in Figure 8.

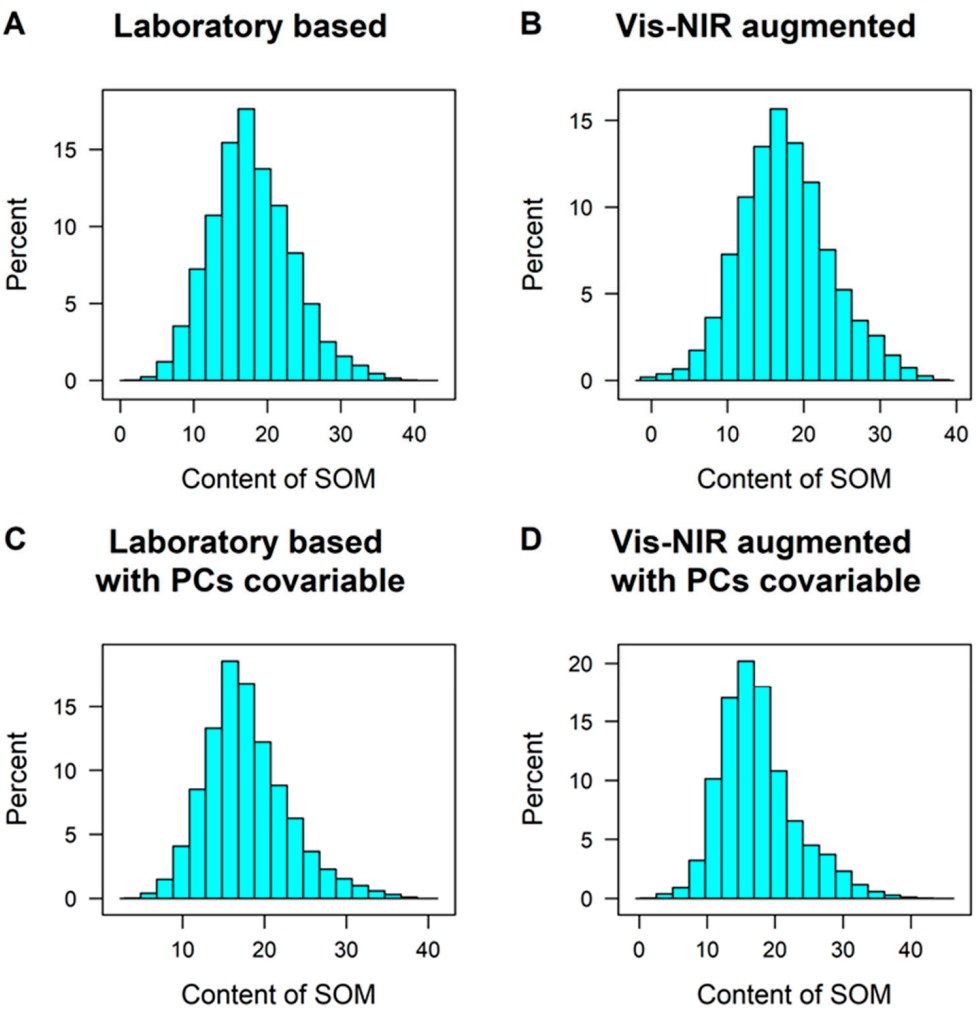

**Figure 8.** Histogram of predicted SOM using the data of the laboratory-based (Lb), vis-NIR augmented datasets (Au.p), the laboratory-based with pcs covariate (Ld.co), vis-NIR augmented datasets with pcs covariate (Au.p.co).

Figure 9A–D shows the location of the difference value of predicted and measured SOM from Lb (A), Au.p (B), Ld.co (C), and Au.p.co (D) from the validation dataset. The range of difference in SOM from Figure 9A, B was more considerable than that in Figure 9C,D. The greatest difference from Figure 9A, B was located in the northwest, where the estimated SOM values were higher than the measured SOM value, but in the north, where estimated SOM values were lower than the measured SOM value. As shown in Figure 9C,D, the largest difference was located in the northwest and southeast, where the estimated SOM value was also lower than the measured SOM value. The difference between measured and estimated SOM values in the middle of the sampling location from Figure 9C,D were lower than that from Figure 9A,B. The difference in the predicted and measured SOM from the vis-NIR model (Figure 9E) was much smaller than that of the predicted and measured SOM from the spatial models (Figure 9A,D). The larger error in vis-NIR spectra in location was not consistent with the spatial models.

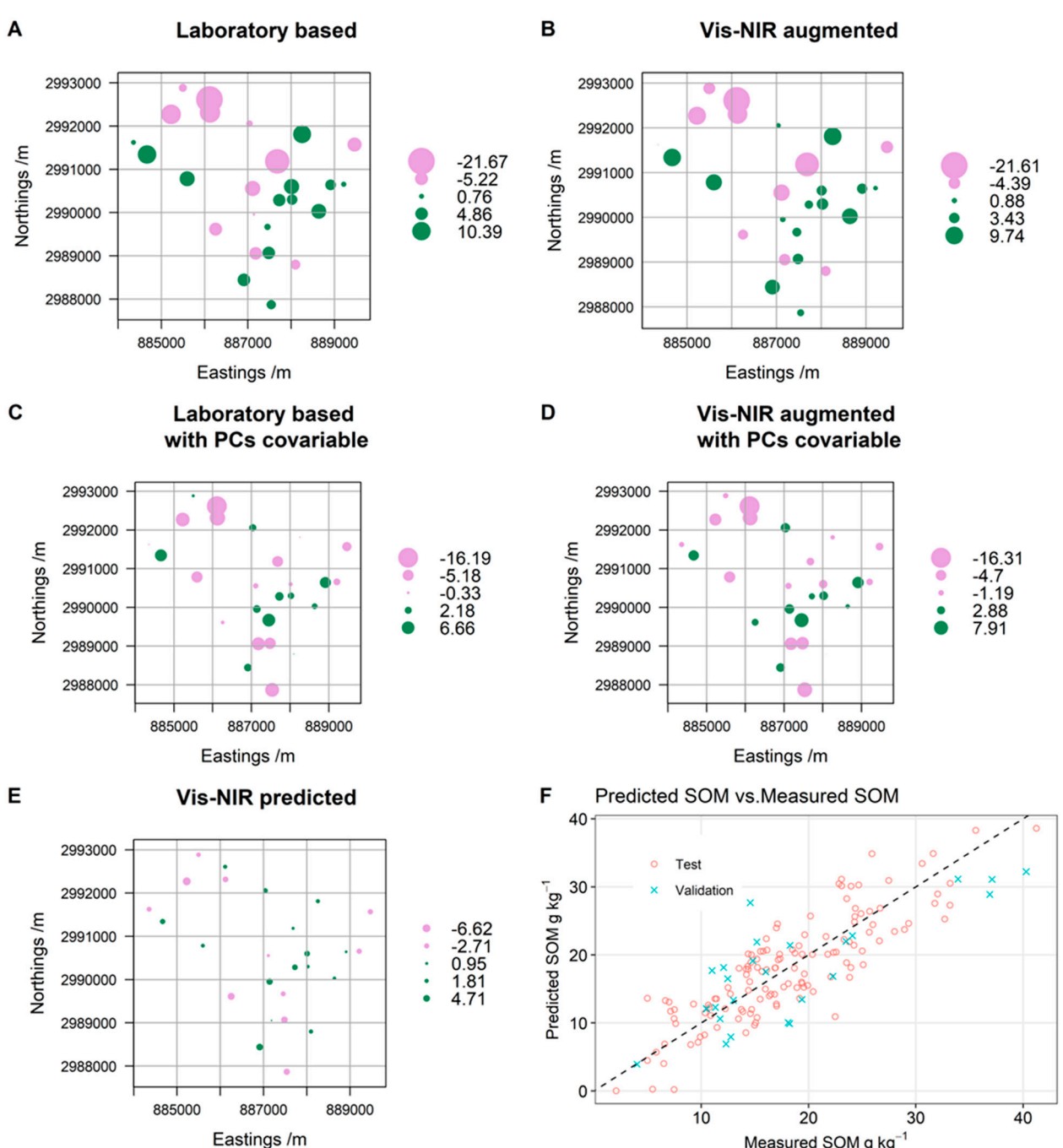

**Figure 9.** The location of the difference value of predicted and measured SOM from Lb (**A**), Au.p (**B**), Ld.co (**C**), and Au.p.co (**D**), and from the vis-NIR Model (**E**) of the validation dataset, the SOM content from the predicted and measured dataset from the test and validation dataset, and the plot of measured SOM and predicted SOM (**F**). The data showed in (**A**)–(**E**) was the difference of the measured SOM minus the predicted SOM.

## 4. Discussion

The difference in SOM with their locations from the validation dataset with the spatial model and from vis-NIR augmented models (Figure 7 and Table 3) showed that the use of visible and near-infrared (vis-NIR) spectroscopy for SOM mapping is an alternative to overcome the time and budget constraints, which can be confirmed by several studies [10,24–26]. However, these researchers only used vis-NIR spectra to predict the properties and did not use vis-NIR spectra as covariates. In this study, we used both the spectra from the test

dataset to predict SOM and PCs of spectra from all samples as covariates. We used PC1 to PC7 as the covariates, which were strongly correlated with SOM. When comparing the prediction from the dataset of laboratory-based (Lb) and vis-NIR augmented with PCs as covariates (Au.p.co), the number of samples used for calibration for spatial data was reduced to nearly 60%, but the accuracy improved by 22%. The main difference between the two maps was that the maps predicted with the vis-NIR augmented model were smoother than those predicted with the laboratory-based model. These can be explained by the fact that the prediction from the vis-NIR model tends to smooth the variation.

Our study showed that the addition of spectroscopy as the covariate was useful in improving the prediction accuracy using kriging. The predicted results using spectroscopy after different preprocessing methods were obviously different (Figures 5 and 6, Table 3). The effect of the covariate was far greater than the effect from the augmented data (Figure 5 and Table 3). Selecting the optimal preprocessing method can be observed from the correlation of spectra with SOM from the calibration dataset (Figure 2B Section B). In this study, lg_dt and la-snv had 1204 and 1248 significant bands, respectively, which were more than the last two. Although msc_snv had the lowest RMSE in the validation when the PC was 5, the prediction in the test dataset was the lowest, which showed that this could not be optimal. The other key in prediction is to select the number of principal components. The correlation of each PC with SOM can be useful (Figure 3A).

Although improved accuracy of maps was produced with the spatial models using the PCs as the covariate, there was still a gap for high-resolution mapping. The difference value from the spatial model from the sampling location showed that the error might be related to the topography; the larger the error is, the larger the value of the DEM (Figure 1), which can be confirmed by Viscarra Rossel et al. [27], who used the DEM as the variable to predict the soil carbon in cool temperate areas. In our study area, the higher the DEM, the lower the temperature, the greater the soil organic matter content, and the larger the difference between the predicted value and measured SOM (Figure 9A–D). Reference [28] also showed using a multi-depth vis-NIR spectral library and terrain attributes in digital soil mapping at the local scale. In this study, we did not consider the DEM due to the small area and the lack of a significant correlation between the DEM and SOM.

The spatial model did not produce a more accurate result than the research of Reference [29], who used representative samples for calibration. In this study, we used random selection methods. The relatively larger RMSE and lower bias from the spatial model showed that imprecision was the main reason [28,30]. The imprecision was due to the large sill value and the larger range of scale from the fitted variogram figure, which showed strong spatial dependence and considerable variation among the SOM [31,32]. Adding sampling points is considered a possible solution because the sampling density from our study was greater (300 × 300 m), which would decrease the precision of the spatial relationship between spectra and soil properties [33] than the need from precision agriculture for soil mapping, which called for one to five samples per hectare [34,35] Therefore, collecting a larger number of soil samples that represent soil spatial variation within the area and measuring their spectra and using them as the covariate in spatial mapping can be a feasible measure for a future study.

We did not investigate the uncertainty of the variogram parameters due to a regular grid sampling design. The estimated values from the nugget variance could not be accurate [29]. A main solution would be to include a second-phase survey which requires many samples. The vis-NIR spectral libraries can be considerable [28,30].

## 5. Conclusions

This study verified that vis-NIR spectroscopy is an alternative to overcome the time and budget constraints of traditional chemical analysis methods in mapping SOM. In this study, we compared the 116 predicted results from the spatial models with the data from the laboratory-based dataset (Ld), augmented dataset predicted using vis-NIR (Au.p), a laboratory-based dataset with vis-NIR spectra as the covariance (Ld.co) and augmented

dataset predicted using vis-NIR with vis-NIR spectra as the covariance (Au.p.co). The conclusions were drawn as follows:

The effect of spectra used as the covariance plays a crucial part in the predicted accuracy. Using different preprocessing methods on spectra had different influences, and the most effective method can be decided by the correlation value from the spectra with SOM from the calibration dataset.

When the vis-NIR model and vis-NIR spectroscopy were used as covariates for SOM mapping, the number used for spatial calibration was reduced to nearly 60%, but the accuracy improved by 23%. The prediction error may be mainly due to the imprecision, not the bias; collecting a larger number of spectra of soil samples that represent soil spatial variation within the area and using them as the covariate in spatial mapping can improve the mapping accuracy. Our conclusion made in this study need to be verified by applying this strategy to spectra measured in a large area.

**Author Contributions:** Conceptualization, M.Y. and X.Z.; methodology, M.Y.; investigation, X.G.; writing—original draft preparation, M.Y.; writing—review and editing, S.C.; supervision, Z.S. and X.Z. All authors have read and agreed to the published version of the manuscript.

**Funding:** This research was supported by grants from the National Natural Science Foundation of China (No. 41061031), the Natural Science Foundation of Jiangxi Province (20212BAB205022), and the Science and Technology Research Project of Jiangxi Provincial Department of Education (No. GJJ181150).

**Data Availability Statement:** The data that support the findings of this study are available from the corresponding author.

**Conflicts of Interest:** The authors declare no conflict of interest.

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
