# Peer review of "Exploring the Potential of vis-NIR Spectroscopy as a Covariate in Soil Organic Matter Mapping"

_remotesensing, doi:10.3390/rs15061617_

Round 1

Reviewer 1 Report (New Reviewer)

Dear Authors,

The paper is well-organized and written. However, the paper has some suggestions to improve. You can find my detailed suggestion in the paper. You should consider them.

Best wishes,

Author Response

Reviewer 1

You should add some information about land cover, agriculture, pasture, forest or etc., of study area.

Response: We had add the texture, soil type and land cover in this study area.

The soil spectra were resampled to 500 to 2450 nm with every 10 spectra to reduce high dimensionality and collinearity. How did you check high dimensionality and collinearity among soil spectra?

Response: Thank you. In this study, we did not use the resampled spectra.

The reflectance percentage is not true. You should revise the left column of the figure.

 Response: thank you. We had corrected it.

Specify in some more detail the wet chemistry method for SOM (lines 126-128). Otassium dichromate methods are base on wet oxydation of organic carbon and NOT organic matter. Probably the literatur you refer to haveconverted SOC into SOM. Please specify this conversion. "

Response: We had revised: SOC content was measured using H2SO4-K2Cr2O7 oxidation method at 180∘C for 5 minutes method according to the methods of the Institute of Soil Science of the Chi-nese Academy of Sciences [22]. The SOM content was obtained using SOC multiplied the coefficient of 1.72 which were suggested by reference [23]

Reviewer 2 Report (Previous Reviewer 2)

After the suggested corrections the manuscript was improved.

There is a typo (line # 170) “components”.

I'm sorry to bother the authors but there are some discrepancies of references in the text i.e.: (line#374) Rizzo et .al is [13] not [14]; and (line# 379) Ramirez-Lopez et al. is [30] not [29].

Please check the bibliography carefully once again

Author Response

There is a typo (line # 170) “components”.

 Response:We had corrected.

I'm sorry to bother the authors but there are some discrepancies of references in the text i.e.: (line#374) Rizzo et .al is [13] not [14]; and (line# 379) Ramirez-Lopez et al. is [30] not [29].

Response: We had corrected.

Please check the bibliography carefully once again

Response: Thank you. We had checked through the bibliography carefully. 

Reviewer 3 Report (Previous Reviewer 4)

Most of my comments have been appropriately incorporated. The manuscript has improved significantly. The introduction still needs to be improved by citing more recent publications and show the novelty of the study in comparison to all of the existing research. The English still needs improvements in some places.

Author Response

Response: Thank you. We added: Another general method in DSM with vis-NIR spectra was to use the predicted value from the nonlinear model (such as Cubist, Random Forest) using spectra and the environmental covariates [16, 17]. These studies can tell us the usefulness of vis-NIR for mapping but did not show how much the extent of vis-NIR can improve. 

Reviewer 4 Report (Previous Reviewer 5)

The authors have revised the manuscript according to the comments, and I think the manuscript in current form can be accepted for publication.

Author Response

Response: Thank you

Reviewer 5 Report (Previous Reviewer 1)

Please, see comments attached.

Author Response

Line 107. The word ‘standardNormalVariate’ should be ‘standard normal variate’

Response: We had revised it.

Figure 1. Remove the north arrow in figure 1a because there are latitude/longitude marks that clearly indicate orientation. Reduce the size of the study site in 1b to match the size of the study site in 1a and adjust the scale in 1b accordingly.

Response: Thank you. We had revised.

Line 133. Define the acronym PC on the first mention.

Response: We had defined.

Lines 326 – 339. Move this paragraph before Figure 9

Response: We had revised.

Lines 329 – 332. I don’t arrive at the same conclusion when looking at figures 9A and 9B. How can the reader identify overestimation and overestimation on the graphs (i.e., larger/smaller circles?).

Response: We added a word to tell the detail. The data showed in A,B,C,D,E was the difference of the measured SOM minus the predicted SOM.

Line 331. Consider replacing the word ‘but’ with ‘while’ or another transition word

Response: We had revised it.

This manuscript is a resubmission of an earlier submission. The following is a list of the peer review reports and author responses from that submission.

Round 1

Reviewer 1 Report

Dear authors, 

Please, see my comments in the attached document.

Reviewer 2 Report

The authors, in this manuscript, propose the Vis-NIR spectroscopy to improve the accuracy in soil mapping and to reduce the cost of measurements, especially for soil organic matter estimation.

The text needs to be revised because there are many inaccuracies.

On the first page line 9 please delete: “Affiliation 1; [email protected]  Affiliation 2; [email protected]

Materials and Methods

What type of chemical analyzes were carried out  to measure the SOM and which method was used?

How many points (bands) had the spectra recorded?

Although it has been explained in the abstract that: the calibration set was used for building the models; the test set was used for generating the augmented data and the validation set was used for evaluating the map accuracy, of which there is no trace in the text.

Which was the level of significance of the covariate PCs? P<0.01 or p<0,05?

Subscript indices “i” are missing in the formula (1)

Pag 4 line 130  the word “components” is missing after principles.

In figure 2 section B, what do the letters (C)  and (D) refer to?

Results

Table 1 -  It is good practice, in order to make correct predictions, that the range of reference values ​​of the calibration set is wider than the test set. I suggest the authors move the sample of the test set, which has the minimum SOM value equal to 2.06, in the calibration set.

Pag 5 line 168 - The authors refer to a Stenberg et al., 2010  study which, however, has no correspondence with the References. The same for Ramirez-Lopez et al. (2019) at line 340.

Prediction of PLSR

How many Latent Variables (LV) were considered in each calibration model?

I don’t understand why “The pre-processing of msc.snv was effective for the validation but not helpful for the test validation” (lines 178-179). Please explain in a better way.

Line 179-180 - like before. I don’t understand why “The ME values showed that predictions by the vis–NIR models for validation were positively biased.”

Correlation analysis.

It is not clear what the significant PCs are. Perhaps it would be better to indicate them in figure 4 A with an asterisk.

The statement:  " However, after msc_snv and lg_sg were proposed, the significant correlation value showed that the PCs were 7. "  is obscure and difficult to interpret.

Lines 193-194 -Does the number of significant bands correspond to the amount of spectra points?

Line 204 - Fig 5 shows that along each row… but along each column…

Figure 6 caption- I think that the red lines show the Bias not the RMSE

Lines 248-250 – “The ME values in Table 3 show that the spatial predictions from the cross-validation were always slightly negatively biased, but the validation was always positively biased.” ??? please explain in a better way.

Lines 268-272- Looking at the Fig. 8, it seems the exact opposite of what the authors said. Please check the figure or the text.

Discussion and Conclusion

Line 310 and line 369- “the number used for spatial calibration”: the number of what?

Line 324 - “the prediction in the test dataset was the lowest”. Does that mean it's worse? And if so why then is it claimed that it could be optimal? Please explain in a better way.

References must be checked carefully: some are missing, others have the wrong number,  i.e. Rizzo et al (2016)

Reviewer 3 Report

The article has scientific relevance and its text is well described in sections adequately distributed. The discussion has a good theoretical foundation updated with important references in pedology and spectroscopy. However, as I will mention, the text needs improvement regarding the introduction and methods.

In the introduction, consider that the public mentioned is just an example of the potential beneficiaries of the information generated in this article (p. 1, l. 39-40). I suggest emphasizing this or including other users in this group, such as public policy managers and environmental and agricultural scientists.

I also suggest writing a paragraph explaining the article's structure at the end of the introduction.

Better clarify the hypothesis and the research problem that needs more argumentation. This point implies in the conclusion.

As for the methodology, it is necessary to detail the methodology: method (auger, trench?) and surface collection depth. Did you collect the samples in triplicate or single? This missing also applies to spectroscopic analysis parameters.

Add a space between words and quotations, for example, SOM[25,26] p. 15, l. 345

Insert space before the lines of the second-order subheadings, as they are close to the text of the previous sections. Strictly follow the journal's guidelines.

I recommend adjusting the position of figures within the text to organize and flow the content; for example, you can refer to these elements in the paragraph above and return to them in the next paragraph.

Figure 1. (p. 2, l. 80) shows a cartographic record problem at the location map. How can the reader know a higher or lower area if he did not access the scale bar with the low and higher values? In addition, they can improve the visual aspect.

Table 1. (p. 5, l. 162) n is the number – what number?

Table 2. (p. 6, l. 182) tables and figures are self-explanatory, so you should describe the acronyms even if you have already described them throughout the text. So I suggest adding a footnote.

The color scale in figure 4 (p. 6, l. 195) is not intuitive, as it confuses positive and negative correlations. I suggest using different colors or a progressive scale from the lowest value to the highest.

Regarding the footnote in Figure 4, you should convert it into a heading starting with a connective like "it is noteworthy that..." and then discuss the implications of your data.

Page numbering is discontinued from page 10.

The text requires adjustments in writing, above all, to avoid the passive voice.

Reviewer 4 Report

The authors predict SOM by vis-nir spectroscopy for a test site in the eastern Jiangxi Province, China. 261 soil samples were taken and analyzed in the laboratory. Four different preprocessing methods were applied before spatial predictions or mapping of SOM was conducted. Kriging was performed to interpolate the spatial prediction and generate a SOM map.

Some major aspects should be improved in the manuscript:

-       The manuscript generally lacks innovation, since quite a lot studies still performed SOM prediction based on vis-nir data. Some of the studies are cited by the authors. Please elaborate on the new aspects and methods provided in this research! What was your ground truth? How were the data used for the prediction?

-       The description of the different datasets is not clear for the reader – Fig. 2 is not straight forward and should be more focused on the methodological workflow with precise definitions and differentiation of the datasets

-       Grammar and spelling of the entire manuscript is not good, thus the reader gets confused by the description of the workflow. Please improve this and provide more details of your methods.

-       What do you mean with augmented data?

-       What is your definition of soil clod? what depth is the soil sample taken from?

-       Provide more information about the spectral measurements: which detection angle, light source and angle

-       Table 4: What is the difference between the rows?

-       Fig. 5/6: Its not clear to me what the figure shows. What is the difference of the datasets and what kind of combinations can be seen here?

-       After clearance of the above points, further review of the results is possible

Various minor aspects:

-       Line 42/43: This is a generalization that does not hold in any case

-       Line 45: an alternative (others exists)

-       Line 46: Do not mix “vis” and “Vis”

-       Fig. 1: Add map of country

-       Line 91: Define w, p, and m

-       Sec 2.4: The DEM shows a very flat and leveled terrain, thus correlations to the DEM were not expected in the dataset. Or not?

-       Define PC (Principal Component)

-       Define significance in your study

-       Eq. 1: Ui is not used in the formula. Please be precise and don’t mix variables and symbols

-       Line 168: Please use uniformly referencing

-       Line 176: pre-processing instead of pro-processing

Reviewer 5 Report

The manuscript “Exploring the potential of vis-NIR spectroscopy as a covariate in soil organic matter mapping” focused on using vis-NIR spectroscopy as a covariate to improve the accuracy of SOM mapping. Generally, the research was well designed and presented, and the topic is interesting and suitable for Remote Sensing. The logic is flow and easy to follow, with intensive work and excellent presentation of results. Thus, I suggest accept the manuscript for publication with minor correction. However, there are still some issues should be focused before the manuscript being accepted for publication.

Comments and Suggestions for Authors

Line 30-31 Here the unit for SOM is g/kg, while in the main text it is g kg-1, please check.

Line 80 For Figure 1B, it is better to use high-resolution satellite images (Sentinel 2 or GE images) as base map rather than DEM. As you have mentioned in Line107-108: We analyzed the correlation of SOM with the terrain attributes (terrain roughness, vector terrain roughness, slope of aspect, slope of slope), but the correlations were quite low, topography is supposed to less affect SOM variability in plain areas. Thus, satellite images may give more detained information about your study area.

Line 195 Fig.4 A, the color scale is confusing, as both positive and negative correlations were illustrated by warm color. It is better to use warm color indicate positive correlations while use cool color for negative relations.

Line 289 I found that most figures are with low resolution (may be over compressed by the word file), and the text in figures (Fig. 1 and Fig. 7) are small. Please check.
